# Identifying optimal first-line immune checkpoint inhibitors based regiments for advanced non-small cell lung cancer without oncogenic driver mutations: A systematic review and network meta-analysis

**Tingting Liu[1,2☺‡], Sijia Wu[3,4☺‡], Weiwei Fang[2☺‡], Hongkai Li[3,4], Lili Su[2], Guiye Qi[5], Huaichen Li[2], Yao Liu[2]***

**1** Shandong University, Jinan, Shandong, China, **2** Department of Respiratory and Critical Care Medicine, Shandong Provincial Hospital Affiliated to Shandong First Medical University, Jinan, Shandong, China, **3** Institute for Medical Dataology, Cheeloo College of Medicine, Shandong University, Jinan, People's Republic of China, **4** Department of Epidemiology and Health Statistics, School of Public Health, Cheeloo College of Medicine, Shandong University, Jinan, People's Republic of China, **5** Department of Medical Engineering Management, Shandong Provincial Hospital Affiliated to Shandong First Medical University, Jinan, Shandong, China

☺ These authors contributed equally to this work.
‡ These authors share first authorship on this work.
* doctorliuyao@126.com

**Data Availability Statement:** The relevant data are within the Supporting Information files.

## Abstract

### Background

Immune checkpoint inhibitors (ICIs) have changed the treatment pattern of advanced and metastatic NSCLC. A series of ICI based therapies have emerged in the first-line treatment field, but the comparative efficacy was unclear.

### Method

We searched multiple databases and abstracts of major conference proceedings up to Apri1, 2022 for phase III randomised trials of advanced driver-gene wild type NSCLC patients receiving first-line therapy. Outcomes analyzed included progression free survival (PFS), overall survival (OS), and et al.

### Results

Thirty-two double-blind RCTs were included, involving 18,656 patients assigned to 22 ICI-based first-line regimens. A series of ICI regiments (including ICI plus chemotherapy), ICI monotherapy, doublet ICIs, doublet ICIs plus chemotherapy) emerged, and showed significant PFS and OS benefit than chemotherapy and chemotherapy + bevacizumab (BEV) for advanced wild-type NSCLC. In comprehensive terms of PFS, chemoimmunotherapy (CIT) were significantly more effective than ICI monotherapy and doublet ICIs. In terms of OS for patients with non-squamous NSCLC, pembrolizumab containing CIT was associated with a

**Funding:** This work was supported by Natural Science Foundation of Shandong Province (No. ZR2021MH006), and the corresponding author doctor LY received this award. The funders of this work had no role in study design, data collection and analysis, decision to publish, or preparation of the manuscript.

**Competing interests:** the authors declare that they have no conflict of interest.

median rank of the best regimens, and followed by Atezolizumab+BEV based CIT; while for OS in patients with squamous NSCLC, Cemiplimab and sintilimab based CIT were the most effective regimens. For more than 2 years follow-up, the atezolizumab, pembrolizumab, nivolumab and durvalumab containing ICI therapy all provide a durable long-term OS benefit over chemotherapy and BEV + chemotherapy.

## Conclusions

The findings of the present NMA represent the most comprehensive evidence, which might suggest or provide basis for first-line ICI therapy decision for advanced NSCLC patients without oncogenic driver mutations.

## Introduction

### Contribution to the field

Immune checkpoint inhibitors (ICIs) including anti–programmed death-1 (PD-1) monoclonal antibody and PD-L1 have greatly changed the treatment pattern of advanced and metastatic NSCLC. Recently, a series of phase 3 trials evaluated the efficacy of new anti-PD-1 antibodies have been published, and new combined ICIs approved indications for advanced NSCLC. At ASCO 2020, impressive long-term PFS and OS data were published with the updated final analysis of KEYNOTE-189 [4] and CheckMate 227. Moreover, at WCLC &ESMO 2021, data of several new first-line phase 3 clinical trials assessed ICI therapy have been published as abstracts. Increasing treatment options may reshape the first-line ICI interventions for advanced NSCLC patients. However, there remains considerable debate about the best therapeutic model (i.e., immunotherapy as single agent or in combination, and how to choose optimal combination therapy), and the potential differences in tolerability between individual treatment strategies. With the marketing of new ICI regiments and increasing numbers of trials published and long-term outcomes updated recently, we conducted an updated network meta-analysis to compare the treatment efficacy of different first-lines regimens for advanced NSCLC patients without oncogenic driver mutations.

Lung cancer remains the leading cause of cancer mortality worldwide, which caused 1.8 million deaths in 2020. Non-small cell lung cancer (NSCLC) accounts for about 80–90% of lung cancer cases [1]. Immune checkpoint inhibitors (ICIs) including anti–programmed death-1 (PD-1) monoclonal antibody and PD-L1 have greatly changed the treatment pattern of advanced and metastatic NSCLC [2–6]. ICI based therapies have been recommended as standard first-line regimens in metastatic or advanced NSCLC patients without oncogenic drivers, as monotherapy or combined with chemotherapy according to PD-L1 expression [7].

Currently, evidence suggests that the OS benefit from monotherapy was mainly observed in patients with high PD-L1 expression (PD-L1 TPS≥50%) in first-line therapies for advanced NSCLC, such as pembrolizumab in KEYNOTE 024, atezolizumab in IMpower 110 or cemiplimab in EMPOWER-Lung [5, 6, 8, 9]. Whereas for patients with PD-L1-intermediate expression (1%≤ PD-L1 < 50%), ICIs + chemotherapy is generally considered the best option. Based on IMpower 150, atezolizumab+ cevacizumab (BEV) + chemotherapy (frequently referred to as ABCP) are also recommended by the US FDA and the European Medicines Agency (EMA) as the first-line treatment of patients without oncogenic drivers [10, 11]. The innovative CheckMate 9LA study demonstrated rapid disease control with limited-course chemotherapy

plus IO doublet, and provided a new first-line treatment option [12]. Additionally, a series of phase 3 trials evaluated the efficacy of new anti-PD-1 antibodies developed by Chinese pharmaceutical companies have been published, and new combined ICIs approved indications for advanced NSCLC by the National Medical Products Administration (NMPA) in china [13–18]. At ASCO 2020, impressive long-term PFS and OS data were published with the updated final analysis of KEYNOTE-189 [4] and CheckMate 227 [19, 20], both regardless PD-L1 expression. Moreover, at WCLC & ESMO 2021, data of several new first-line phase 3 trials analyzed the efficacy of ICI therapies were published as abstracts [21, 22]. Increasing treatment options may reshape the first-line ICI therapeutic strategy for advanced NSCLC.

There remains a lot of controversy about the best therapeutic model (i.e., immunotherapy as single agent or in combination, and how to choose optimal combination therapy), and the differences in treatment-related adverse events between individual regiments. With the marketing of new ICI regiments, the most recent clinical trials findings, and long-term outcomes updated recently [4, 19–22], we conducted an updated network meta-analysis to compare the different first-lines regimens in term of short and long term PFS and OS for advanced NSCLC patients without oncogenic driver mutations.

## Methods

### Ethical approval

Ethics approval was obtained from the Ethics Committee of Shandong Provincial Hospital affiliated with Shandong First Medical University.

### Search strategy and selection criteria

This study is registered with PROSPERO, number CRD42021291015, and is reported according to the Preferred Reporting Items for Systematic Reviews and Meta-Analyses (PRISMA) extension statement for network meta-analysis.

We searched PubMed, the Cochrane Library, Embase, the World Health Organization (WHO) International Clinical Trials Registry Platform (ICTRP), including conference proceedings of the World Conference on Lung Cancer (WCLC), the American Society of Clinical Oncology (ASCO), the European Society of Medical Oncology (EMSO), and the Chinese Society of Clinical Oncology (CSCO) Academic Annual Conference, from inception until Apri1, 2022, with no language restrictions. The following Medical Subject Headings were used to search: non-small cell lung cancer (including non-squamous lung cancer, squamous lung cancer), and immunotherapy (including all currently known ICIs: pembrolizumab, atezolizumab, ipilimumab, nivolumab, avelumab, sintilimab, durvalumab, tremelimumab, tislelizumab, toripalimab, sugemalimab, camrelizumab, and et al.). Reference lists of relevant studies were also screened.

The inclusion criteria were as follows: studies that analyzing the efficacy and safety of ICIs (ICI monotherapy or double ICIs) alone or combined with chemotherapy and or antiangiogenetic drugs, as first-line treatments for advanced NSCLC; and reports contained patients' data, complete protocol, and at least one of key clinical outcomes, such as progression-free survival (PFS), overall survival (OS), and objective response rate (ORR), as well as the incidence of treatment-related adverse events (TRAEs). Phase III trials that analyzing the efficacy of chemotherapy in combination with BEV or placebo as first-line therapy for nonsquamous NSCLC were also included for comparison and to form a connected network.

The following studies were excluded from this review: those involving previously treated patients with advanced NSCLC; studies involving operable NSCLC who were treated with ICI therapy before surgery or after surgery; studies not reported key clinical outcomes (PFS, OS,

ORR, the incidence of TRAEs); and studies lacking valid data for evaluating the safety and effi-
cacy of ICI first-line treatment.

## Data extraction

Two investigators (YL and WF) independently examined the titles and abstracts of retrieved
articles to assess the eligibility. The full articles were evaluated if a decision could not be made
based on the titles and abstracts. Data were extracted by the same two reviewers (YL and WF)
using a predefined spread sheet. The extracted data included: trial name, year of publication,
study design, number of participants, median/mean age, percent males, histology type, disease
stage, PD-L1 expression level, median duration of follow-up, and outcome data. Outcome data
including PFS, OS, and ORR were extracted; TRAEs associated with each intervention were
also extracted in this network meta-analysis (S1 Table).

The full texts screen was performed, when a decision could not determine based on the
titles and abstracts. Two pairs of investigators (YL, WF, SWM ans SLL) extracted data from
the studies or from supplementary materials, and evaluated the risk of bias. Disagreements
were resolved through discussion.

We use the data of PFS, and ORR assessed by blinded independent central review (BICR)
or independent radiographic review committee (IRRC), if available. Otherwise, investigator-
assessed PFS, and ORR were used. Hazard ratio (HRs) and 95% confidence intervals (CIs) for
PFS and OS were extracted, percentage for long-term PFS and OS rate were also extracted.
Likewise, dichotomous ORR data, grade $\geq 3$ AEs and immune-mediated were clustered.

## Evidence networks

We performed fixed-effects network meta-analyses (NMA) in our study.

According to trial characteristics, heterogeneity analysis was performed on all eligible stud-
ies. The NMA meets the technical requirement that each treatment should be represented by
at least one clinical study to create a viable comparison network. Under the assumption of con-
sistency, the NMA model associated data from the individual studies with basic parameters
that reflect the (pooled) relative treatment effect of each intervention compared with the refer-
ence therapy. Based on these parameters, the relative treatment effects of each contrast treat-
ment in the network were obtained. Under the Bayesian framework, we could calculate the
probability of being the best treatment out of all treatments included in the connected net-
work. It is also possible to calculate which of all the interventions included in the connected
network are the best and which are the next best. The 'rank probability' function was per-
formed to calculate rank-probabilities, for each MCMC iteration, the treatments are ranked
according to their effect relative to baseline. A frequency table was constructed according to
these rankings and normalized by the number of iterations to give the rank probabilities.

Three histology-based evidence networks were constructed in our analysis: a mixed-histol-
ogy network, a squamous NSCLC network and a non-squamous NSCLC network. The results
of the NMA for PFS and OS were presented with estimates of the treatment effects of each
intervention with respect to the reference therapy, chemotherapy. The posterior distributions
of relative treatment effects were performed by the median and 95% CIs, which were made up
of the 2.5th and 97.5th percentiles of the posterior distributions. If the 95% CIs were
completely below or above 1, the respective estimated ratios were considered significant. Addi-
tionally, we used SUCRA to perform the rank of all treatments, the closer the SUCRA value is
to 1, the better the treatment effect is.

Additional subgroups analyses were performed according to histology (i.e., squamous/non-
squamous) and levels of PD-L1 expression (i.e., PD-L1-high: PD-L1 $\geq 50\%$; PD-L1-negative:

PD-L1 < 1%; PD-L1-intermediate: 50%>PD-L1 ≥ 1%). Statistical analysis was carried out using the 'gemtc' version 1.0–1 of R-4.1.1 software.

## Results

### Search results

A total of 2,375 citations were identified by the comprehensive search strategy (Fig 1). After excluding duplicate citations, 1765 citations underwent title/abstract screening and 65 studies

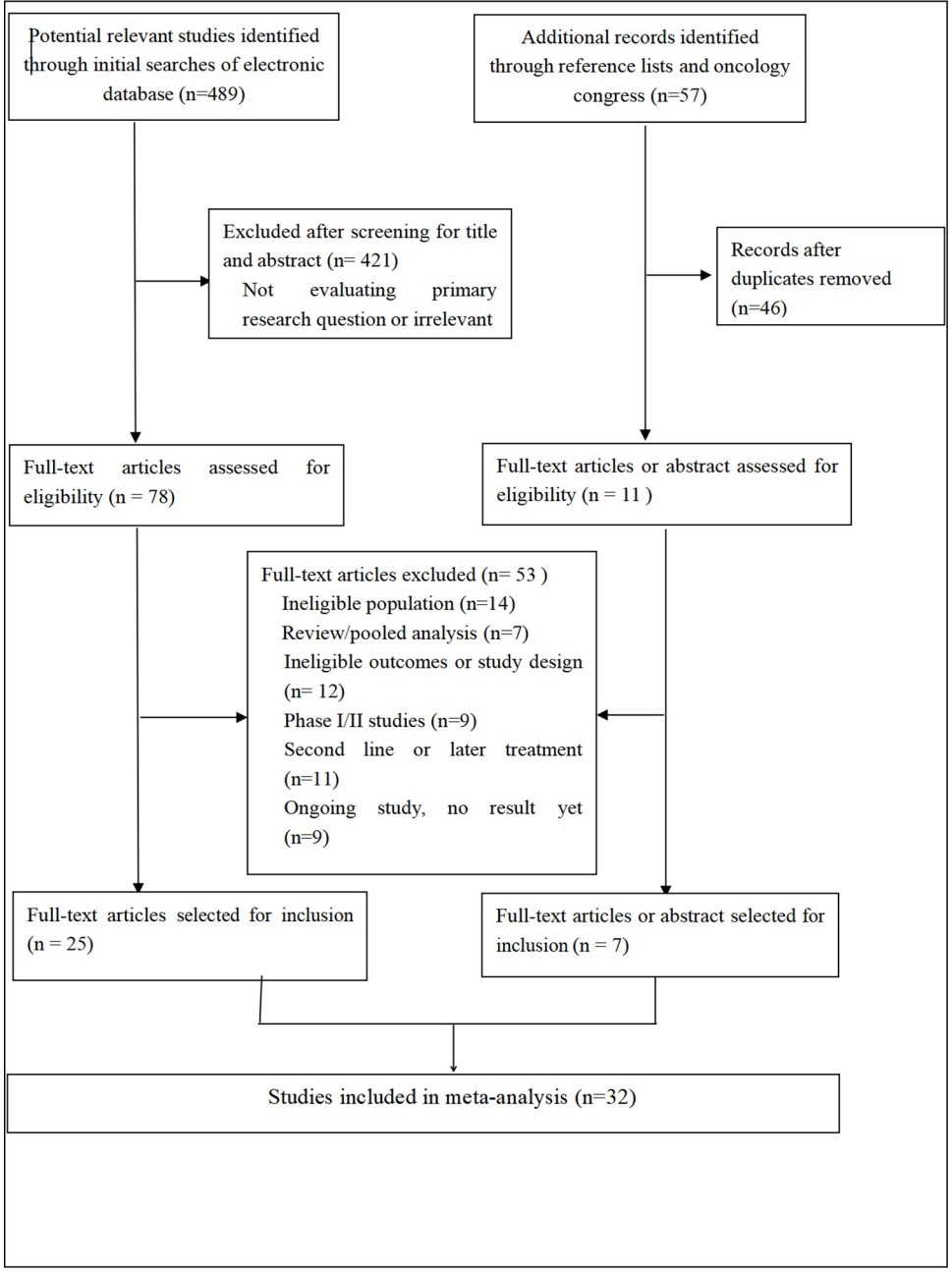

**Fig 1. PRISMA flow diagram (up to April 1, 2022).**

were retrieved for full text screen. Of the 69 full-text articles screened, 37 studies not meeting the inclusion criteria were excluded. Overall, 32 first-line phase III RCTs (comprising 18,656 participants) done between 2016 and 2022 were included in the analysis. Study characteristics of included trials are listed in Table 1. The mean sample size was range from 1274 to 305 participants.

## Trial characteristics

All 32 trials were randomized controlled phase 3 trials with an open-label (17 studies) or a double-blind (15 studies) trial design [4–6, 8–48]. Most of the trials had low risk of bias on the basis of the Cochrane's tool for randomized trials, and the majority of participants with ECOG performance scores (PS) of 0 or 1. All the RCTs included was completed in the recent 5 years. The majority of included trials (22 of 32 studies) were global, multicentre designed trials [4–6, 8–12, 19–21, 23–34, 37–40, 42, 45–48], and 10 trials were multicentre trials conducted in China and reported (or were assumed) to study 100% Chinese patients [13–18, 22, 35, 36, 41, 43, 44].

## Outcomes of interest

PFS and OS were co-primary endpoints in most of the trials included. Similar definitions of OS were used across trials, defined as the time from randomization to death from any reason. In majority trials, PFS was defined as the time from randomization to disease progression or death from any cause, as assessed by the IRRC according to Response Evaluation Criteria in Solid Tumors (RECIST) version 1.1. Whereas, the series of IMpower trials only reported investigator assessed PFS, ORR and DOR [8, 10, 11, 31–36]. In the present analysis, investigator and IRRC assessed outcomes were assumed to be comparable.

## Population

The majority of the trials (16/32) included stage IV or recurrent NSCLC [4, 8, 10–12, 19, 20, 23–25, 27–34, 37–41], and 16 trials included both stage III and stage IV NSCLC [5, 6, 9, 13–18, 21, 22, 26, 35, 36, 42–48]. The randomized participants of the majority trials were patients who had no sensitizing EGFR or ALK genetic alterations. IMpower130, IMpower131, and IMpower150 included patients with EGFR or ALK genetic alterations but required these patients to have progressed from treatment with tyrosine kinase inhibitor [10, 11, 32, 33]. In the present network meta-analysis, patients with confirmed or known ALK/EGFR genetic alterations were excluded. Twenty-six studies randomized NSCLC patients irrelevant of PD-L1 expression [4, 10–25, 29, 30, 32–36, 38–48]. In 3 trials (Keynote-042, IMpower110, CheckMate 026), only PD-L1-positive patients (PD-L1 tumour proportional score (TPS) ≥1% [8, 27, 31, 37] and in the other 3 trials (Keynote-024, Keynote-598, EMPOWER-Lung) only participants with high expression of PD-L1 (TPS ≥ 50%) were eligible for inclusion [5, 6, 9, 26, 28]. For histology status, 14 trials were conducted in both squamous and nonsquamous patients [5, 6, 8, 9, 12, 19–23, 26–28, 31, 37–39, 41, 42], 11 trials only including nonsquamous patients [4, 10, 11, 13, 14, 16, 18, 24, 25, 32, 34, 44–48] and 7 trials entrolled squamous NSCLC patients [15, 17, 29, 30, 33, 35, 36, 40, 43].

The control arm was platinum-based chemotherapy in 31 trials of the included studies, with only one trial (Impover150) added BEV in both the experimental and control arm [10, 11]. Seven trials evaluated the efficacy of ICI-monotherapy, including pembrolizumab, atelizumab, nivolumab, cemiplimab or durvalumab monotherapy [5, 6, 8, 9, 26, 27, 31, 37–39, 42]; 3 trials tested doublet ICIs therapy including pembrolizumab/ipilimumab, nivolumab/ipilimumab, and durvalumab/tremelimumab [19, 20, 28, 38, 39]. Moreover, ICIs in combination with

Table 1. Study characteristics.

| Study | Years | Study design | Sample size | Median ages (years) | Male/ female | Histology | Disease stage | PD-L1 expression | Therapeutic regimen | ChT therapy Drug | Follow-up (month) |
|---|---|---|---|---|---|---|---|---|---|---|---|
| Keynote-189 [4, 24, 25] | 2018, 2020 | double blind, phase III | 410/206 | 65/64 | 363/ 253 | non-squ | IV | All | PEMB + ChT vs. ChT | CAB (AUC = 6)/3w or CIS (75mg/ m²/3w) +PEM (500mg/m²/3w) | 23.1 |
| Keynote-024 [5, 6, 26] | 2016, 2019 | open-label, phase III | 154/151 | 64.5/63 | 187/ 118 | squ/non-squ | IIIB-IV | ≥50% | PEMB vs. ChT | CIS (75mg/ m²/3w) or CAB (AUC = 5–6)/3w +PEM (500mg/m²/3w) | 25.2 |
| | | | | | | | | | | or PTX (200 mg/m² Q3W) or | |
| | | | | | | | | | | GEM (1250 mg/m2 d1,8 Q3W) | |
| Keynote-042 [27] | 2019 | open-label, phase III | 637/637 | 63/64 | 902/ 372 | squ/non-squ | IV | ≥1% | PEMB vs. ChT | 1) CAB (AUC = 5–6)/ 3w+ PEM (500mg/m²) /3w or PTX (200 mg/ m² Q3W) | 12.8 |
| Keynote-598 [28] | 2020 | double blind, phase III | 284/284 | 64/65 | 393/ 175 | squ/non-squ | IV | ≥50% | PEMB+IPIL vs. PEMB+placebo | | 20.6 |
| Keynote-407 [29, 30] | 2018, 2020 | double blind, phase III | 278/281 | 65/65 | 455/ 104 | squ | IV | All | PEMB+ ChT vs. ChT | CAB (AUC = 6, d1)/ 3w+PTX | 14.3 |
| | | | | | | | | | | (200mg/m²/3w d1) or nab- PTX (100mg/m²/ 3w d1, 8, 15), 4 circle | |
| IMpower110 [8, 31] | 2020, 2021 | open-label, phase III | 277/277 | NG/NG | NG/ NG | squ/non-squ | IV | ≥1% | ATEZ vs. ChT | CIS (75mg/ m²/3w) or CAB (AUC = 6)/3w +PEM (500mg/m²/3w) or GEM (1000 mg/m2 d1,8 of Q3W) | 15.7 |
| IMpower130 [32] | 2019 | open-label, phase III | 483/240 | 64/65 | 415/ 309 | non-squ | IV | All | ATEZ + ChT vs. ChT | CAB (AUC = 6)/3w +nab-PTX (100mg/ m²/w), 4 or 6C | 19.2 |
| IMpower131 [33] | 2020 | open-label, phase III | 343/340 | 65/65 | 557/ 126 | squ | IV | All | ATEZ+ ChT vs. ChT | CAB (AUC = 6)/3w+ +PTX (200mg/m²/3w d1) or nab-PTX (100mg/m²/w), 4 or 6C | 18.1 |
| IMpower132 [34] | 2020 | open-label, phase III | 292/286 | 64/63 | 384/ 194 | non-squ | IV | All | ATEZ + ChT vs. ChT | CAB (AUC = 6)/3w or CIS (75mg/ m²/3w) +PEM (500mg/m²/3w) | 14.8 |
| IMpower150 [10, 11] | 2018, 2021 | open-label, phase III | 359/338 | 63/63 | 425/ 267 | non-squ | IV | All | ATEZ +BEV + ChT or ATEZ + ChT vs. BEV +ChT | CAB (AUC = 6)/3w +PTX (200mg/m²/3w) +BEVA (15mg/kg/3w), | 15.5 |
| Camel [18] | 2020 | double blind, phase III | 205/207 | 59/61 | 295/ 117 | non-squ | IIIB-IV | All | CAMR+ ChT vs. ChT | CAB (AUC = 5–6)/3w +PEM (500mg/m²/3w) | 11.9 |
| Camel-sq [35, 36] | 2021, 2022 | double blind, phase III | 193/196 | 64/62 | 359/30 | squ | IIIB-IV | All | CAMR+ ChT vs. ChT | CAB (AUC = 5)/3w+ +PTX (175mg/m²/3w d1), 4 or 6C | |
| CheckMate 026 [37] | 2017 | open-label, phase III | 271/270 | 63/65 | 332/ 209 | squ/non-squ | IV or Recurrent | ≥1% | NIVO vs. ChT | Pd-ChT/3w | 13.7 |
| CheckMate 227 [19, 20] | 2018 | open-label, phase III | 139/160 | 64/64 | 204/95 | squ/non-squ | IV or Recurrent | All | NIVO + IPIL vs. ChT | Pd-ChT/3w | 11.2 |

(*Continued*)

**Table 1.** (Continued)

| Study | Years | Study design | Sample size | Median ages (years) | Male/ female | Histology | Disease stage | PD-L1 expression | Therapeutic regimen | ChT therapy Drug | Follow-up (month) |
|---|---|---|---|---|---|---|---|---|---|---|---|
| EMPOWER-Lung I [9] | 2021 | open-label, phase III | 356/354 | 63/64 | 606/104 | squ/non-squ | IIIB-IV | ≥50% | CEMI vs. ChT | Pd-ChT/3w | 10.9 |
| CheckMate 9LA [12] | 2021 | open-label, phase III | 503/216 | 65/65 | 361/358 | squ/non-squ | IV or recurrent | All | NIVO plus IPIL + ChT vs. ChT | Pd-ChT/3w | 13.2 |
| MYSTIC [38, 39] | 2020 | open-label, phase III | 374/372 | 65/64 | 506/240 | squ/non-squ | IV | All | DURV vs. ChT | Pd-ChT/3w | 30.2 |
|  |  |  | 372/372 | 66/64 | 516/228 | squ/non-squ | IV | All | DURV +TREM vs. ChT | Pd-ChT/3w |  |
| Ipilimumab III [40] | 2018 | double blind, phase III | 388/361 | 64/64 | 635/114 | squ | IV or recurrent | All | IPIL + ChT vs. ChT | PTX (175mg/m²/3w) +CAB (AUC = 6)/3w | 12.5 |
| ORIENT-11 [13, 14] | 2020, 2022 | double blind, phase III | 266/131 | 61/61 | 303/94 | non-squ | IIIB-IV | All | SINT+ ChT vs. ChT | PEM (500mg/m²/3w) +CAB (AUC = 6)/3w or CIS (75mg/ m²/3w) | 8.9 |
| ORIENT-12 [15] | 2021 | double blind, phase III | 179/178 | 64/62 | 327/30 | squ | IIB-IV | All | SINT+ ChT vs. ChT | GEM (1250 mg/m2 d1,8 of Q3W) +CAB (AUC = 5)/3w or CIS (75mg/ m²/3w) | 12.9 |
| RATIONALE-304 [16] | 2021 | open-label, phase III | 222/110 | 60/61 | 247/87 | non-squ | IIIB, IV | All | TISL+ ChT vs. ChT | PEM (500mg/m²/3w) +CAB (AUC = 5)/3w or CIS (75mg/ m²/3w) | 9.8 |
| RATIONALE-307 [17] | 2021 | open-label, phase III | 139/121 | 60/61 | 330/30 | squ | IIIB, IV | All | TISL+ ChT vs. ChT | nab-PTX (100 mg/m2, days 1, 8, and 15) or PTX (175mg/m2) and CAB (AUC of 5) | 8.6 |
| Gemstone-302 [41] | 2020 | double blind, phase III | 320/159 | 62/64 | 383/96 | squ/non-squ | IV | All | SUGE+ ChT vs. ChT | PEM (500mg/m²/3w) or PTX (175mg/m²/3w) +CAB (AUC = 5)/3w | 8.6 |
| BFAST [42] | 2021 | Cohort C | 234/237 | NA | NA | squ/non-squ | IIIB, IV | bTMB≥10 | ATEZ vs. ChT | Pd-ChT/3w | 18.2 |
| EMPOWER-Lung III [21] | 2021 | double blind, phase III | 312/154 | NA | NA | squ/non-squ | IIIB-IV | All | CEMI + ChT vs. ChT | Pd-ChT/3w |  |
| CHOICE-01 [22] | 2021 | double blind, phase III | 320/159 | 63/61 | 377/102 | squ/non-squ | IIIB-IV | All | TORI+ChT vs. ChT | PEM (500mg/m²/3w) or nab-PTX (100mg/m²/w) +CAB (AUC = 5)/3w or CIS (75mg/ m²/3w) |  |
| AK105 [43] | 2021 | double blind, phase III | 175/175 | NA | NA | squ | IIIB/IIIC, IV | All | PENP+ChT vs. ChT | PTX (200mg/m²/3w d1) +CAB (AUC = 5)/3w |  |
| POSEIDON [23] | 2021 | open-label, phase III | 338/337 | NA | NA | squ/non-squ | IV | All | DURV+ChT vs. ChT | Pd-ChT/3w |  |
|  |  |  | 338/337 | NA | NA | squ/non-squ | IV | All | DURV + TREM +ChT vs. ChT | Pd-ChT/3w |  |

(*Continued*)

**Table 1.** (Continued)

| Study | Years | Study design | Sample size | Median ages (years) | Male/ female | Histology | Disease stage | PD-L1 expression | Therapeutic regimen | ChT therapy Drug | Follow-up (month) |
|---|---|---|---|---|---|---|---|---|---|---|---|
| BEYOND [44] | 2015 | double blind, phase III | 138/138 | 57/56 | 152/ 124 | non-squ | IIIB, IV or recurrent | All | BEV+ ChT vs. ChT | CAB (AUC = 6)/3w +PTX (200mg/m$^2$/3w d1) or nab-PTX (100mg/m$^2$/w), 4 or 6C | |
| ECOG 4599 [45] | 2006 | double blind, phase III | 444/434 | NA | 463/ 387 | non-squ | IIIB, IV or recurrent | All | BEV+ ChT vs. ChT | CAB (AUC = 6)/3w +PTX (200mg/m$^2$/3w d1), 6C | |
| PRONOUNCE [46] | 2015 | open-label, phase III | 182/179 | 66/65 | 209/ 152 | non-squ | IIIB, IV or recurrent | All | BEV+ ChT vs. ChT | CAB (AUC = 6)/3w +PTX (200mg/m$^2$/3w d1) or PEM (500mg/ m$^2$/3w), 4C | |
| AVAiL [47, 48] | 2009, 2010 | double blind, phase III | 696/347 | 58/59 | 442/ 223 | non-squ | IIIB, IV or recurrent | All | BEV + ChT vs. ChT | GEM (1250 mg/m2 d1,8 of Q3W) + CIS (75mg/ m$^2$/3w) | |

Data are expressed as intervention/control unless indicated otherwise.

ATEZ, atezilumab; BEV, bevacizumab; CAB, carboplatin; CAMR, camrelizumab; CEMI, cemiplimab; ChT, Chemotherapy; CIS, cisplatin; DURV, durvalumab; GEM, gemcitabine; IPIL, ipilimumab; NIVO, nivolumab; Squ, squamous; Non-squ, non-squamous; Pb-ChT, platinum-based doublet ChT; PEM, pemetrexed; PEMB, pembrolizumab; PENP, Penpulimab; PTX, paclitaxel; SINT, sintilimab; Squ, squamous; SUGE, sugemalimab; TISL, Tislelizumab; TORI, Toripalimab; TREM, tremelimumab.

chemotherapy were evaluated in 20 studies (pembrolizumab:2 studies; ipilimumab:1 study; atezolimumab:4 studies; camrelizumab:2 studies; sintilimab:2 studies; tislelizumab:2 studies; sugemalimab:1 study; cemiplimab:1 study; toripalimab:1 study; durvalumab:1 study; and pen-pulimab:1 study) [4, 13–18, 21, 22, 24, 25, 29, 30, 32–36, 40, 41, 43–48]. Finally, the CheckMate 9LA study evaluated doublet ICIs (nivolumab/ipilimumab) combined with 2 cycles chemo-therapy [12], and the POSEIDON study investigated durvalumab/tremelimumab plus chemo-therapy [23]. The median of follow-up ranges from 8.6 to 59.9 months. Four first-Line trials of BEV plus chemotherapy versus chemotherapy in patients with nonsquamous NSCLC were also included to form a connected network [44–48].

## Outcomes

All 32 trials included in the NMA reported PFS results across 22 different first-line immuno-therapy regimens. Fig 2 and S1 Fig show the full findings of our network meta-analysis for the PFS, OS and ORR outcome.

## Progression-free survival

**PFS-NMA for overall study cohort.** In the overall PFS-NMA, the penpulimab (HR, 0.40 [0.29–0.55]), atezolimumab+bevacizumab (HR,0.44 [0.37–0.53]), camrelizumab (HR,0.45 [0.38–0.55]), sintilimab (HR,0.51[0.42–0.62]), pembrolizumab (HR,0.52 [0.45–0.60]), sugemalimab (HR,0.54 [0.41–0.70]), toripalimab (HR, 0.56 [0.42–0.74]), cemiplimab (HR, 0.56 [0.44–0.71]), tislelizumab (HR, 0.58 [0.46–0.74]), atezolizumab (HR, 0.65 [0.59–0.72]), and nivolumab+-ipilimumab (HR, 0.70 [0.57–0.86]), as well as durvalumab+tremelimumab (HR, 0.72 [0.60–0.86]), durvalumab (HR, 0.74 [0.62–0.88]) based CIT all showed a significant benefit in PFS over chemotherapy. Significant benefit in PFS is also observed for atezolizumab monotherapy (HR, 0.72 [0.67–0.79]), and nivolumab/ipilimumab combined regiment (HR, 0.83 [0.72–

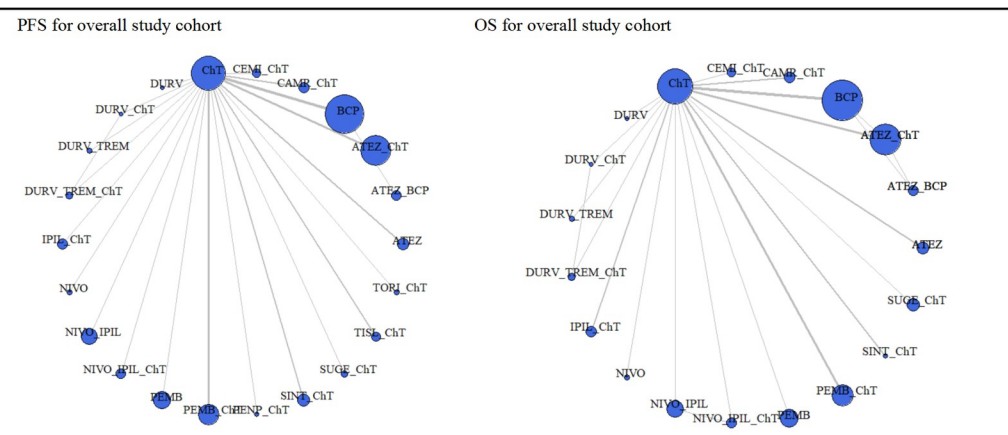

**Fig 2.** Network meta-analysis (NMA) of comparisons for PFS (A) and OS (B). The circle size was corresponding to the participants number (the circle of the chemotherapy group proportional to the number of participants divided by 6). The directly compared regimens are linked with a line, and the thickness of the lines is proportional to the number of trials that assessed the comparison. ATEZ, atelizumab; BEV, bevacizumab; CAMR, camrelizumab; CEMI, cemiplimab; CHT, chemotherapy; DURV, durvalumab; IPIL, ipilimumab; NIVO, nivolumab; PEMB, pembrolizumab; PENP, Penpulimab; SINT, sintilimab; SUGE, sugemalimab; TISL, Tislelizumab; TORI, Toripalimab; TREM, tremelimumab.

0.96]). The penpulimab based CIT showed numerical superiority over other ICI regimens for PFS, which is 95.7% most likely to be the best regiment for PFS, followed by atelizumab + bevacizumab + chemotherapy (ABCP) (92.0%), camrelizumab containing CIT (90.2%) and sintilimab containing CIT (80.4%). These strategies were all significantly more effective than ICI monotherapy and doublet ICIs therapy, and also superior to atelizumab, durvalumab, durvalumab+tremelimumab, and nivolumab+ipilimumab based CIT (P<0.05). No significant difference was founded comparing with penpulimab, atelizumab+bevacizumab, camrelizumab, pembrolizumab, sugemalimab, tislelizumab, toripalimab, or sintilimab based CIT (P>0.05) (Fig 3 and S2 Table).

**PFS-NMA for squamous NSCLC.** In the squamous-histology PFS-NMA, PFS was significantly improved with CIT regiments including sugemalimab (HR, 0.33 [0.22–0.50]), camrelizumab (HR, 0.37 [0.29–0.47]), penpulimab (HR,0.40 [0.29–0.54]), tislelizumab (HR,0.52 [0.37–0.74]), sintilimab (HR,0.53 [0.41–0.68]), cemiplimab (HR, 0.56 [0.40–0.79]), pembrolizumab (HR, 0.56 [0.45–0.70]), and atezolizumab (HR, 0.71 [0.60–0.84]) based CIT, as well as pembrolizumab (HR, 0.35 [0.17–0.71]) and cemiplimab monotherapy (HR, 0.48 [0.34–0.67]). Sugemalimab based CIT is 91.2% most likely to be the best regiment for PFS in squamous NSCLC, followed by camrelizumab containing CIT (86.9%) and pembrolizumab monotherapy (83.8%). Sugemalimab and camrelizumab based CIT were both significantly more effective than pembrolizumab and atezolizumab based CIT for patients with squamous histology (P<0.05); but no significant benefit for PFS was observed when comparing with penpulimab, cemiplimab, sintilimab, or tislelizumab based CIT, as well as pembrolizumab, or cemiplimab monotherapy (P>0.05) (Fig 3 and S3 Table).

**PFS-NMA for non-squamous NSCLC.** In the nonsquamous histology PFS-NMA, all CIT regiments, single-agent ICI (pembrolizumab (HR, 0.55 [0.39–0.77]), nivolumab+ ipilimumab (HR, 0.55 [0.38–0.80]) and cemiplimab (HR, 0.60 [0.44–0.81]) and atezolizumab (HR, 0.65 [0.48–0.88]), and consistently improved PFS, compared with chemotherapy. The addition of Bev in atezolizumab containing CIT (ABCP) is 92.3% most likely to be the best regiment for PFS in nonsquamous NSCLC, followed by sintilimab containing (81.3%) and pembrolizumab containing CIT (79.2%). ABCP performed significantly better than ACP and sugemalimab

### PFS for overall study cohort

| Treatment VS ChT | | HR 95%CI |
|---|---|---|
| PENP+ChT | | 0.4 (0.29,0.55) |
| ATEZ+BCP | | 0.44 (0.29,0.55) |
| CAMR+ChT | | 0.45 (0.38,0.55) |
| SINT+ChT | | 0.51 (0.42,0.62) |
| PEMB+ChT | | 0.52 (0.45,0.6) |
| SUGE+ChT | | 0.54 (0.41,0.7) |
| CEMI+ChT | | 0.56 (0.44,0.71) |
| TORI+ChT | | 0.56 (0.42,0.74) |
| TISL+ChT | | 0.58 (0.46,0.74) |
| ATEZ+ChT | | 0.65 (0.59,0.72) |
| NIVO+IPIL+ChT | | 0.7 (0.57,0.86) |
| BCP | | 0.71 (0.67,0.77) |
| ATEZ | | 0.72 (0.67,0.79) |
| DURV+TREM+ChT | | 0.72 (0.6,0.86) |
| DURV+ChT | | 0.74 (0.62,0.88) |
| NIVO+IPIL | | 0.83 (0.72,0.96) |
| DURV | | 0.87 (0.59,1.3) |
| IPIL+ChT | | 0.87 (0.75,1) |
| DURV+TREM | | 1.1 (0.72,1.5) |
| PEMB | | 1.1 (0.94,1.2) |
| NIVO | | 1.2 (0.91,1.5) |

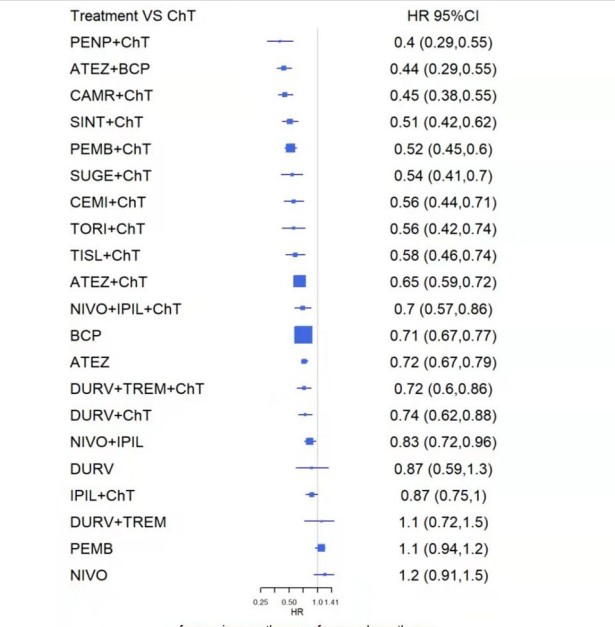

favours immunotherapy  favours chemotherapy

### PFS for non-squamous NSCLC

| Treatment VS ChT | | HR 95%CI |
|---|---|---|
| ATEZ+BCP | | 0.44(0.37,0.53) |
| SINT+ChT | | 0.48(0.36,0.64) |
| PEMB+ChT | | 0.5(0.42,0.59) |
| CEMI+ChT | | 0.53(0.39,0.73) |
| NIVO+IPIL | | 0.55(0.38,0.80) |
| PEMB | | 0.55(0.39,0.77) |
| CAMR+ChT | | 0.6(0.45,0.79) |
| CEMI | | 0.6(0.44,0.81) |
| ATEZ+ChT | | 0.62(0.55,0.71) |
| ATEZ | | 0.65(0.48,0.88) |
| TISL+ChT | | 0.65(0.46,0.90) |
| SUGE+ChT | | 0.66(0.48,0.91) |
| BCP | | 0.71(0.67,0.77) |
| NIVO | | 1.3(1.00,1.60) |

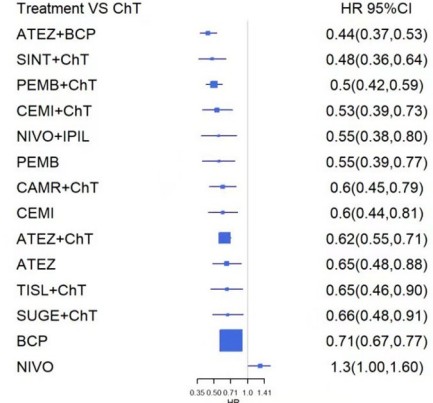

favours immunotherapy  favours chemotherapy

### PFS for squamous NSCLC

| Treatment VS ChT | | HR 95%CI |
|---|---|---|
| SUGE+ChT | | 0.33(0.22,0.50) |
| PEMB | | 0.35(0.17,0.71) |
| CAMR+ChT | | 0.37(0.29,0.54) |
| PENP+ChT | | 0.4(0.29,0.54) |
| CEMI | | 0.48(0.34,0.67) |
| TISL+ChT | | 0.52(0.37,0.74) |
| SINT+ChT | | 0.53(0.41,0.68) |
| CEMI+ChT | | 0.56(0.4,0.79) |
| PEMB+ChT | | 0.56(0.45,0.70) |
| NIVO+IPIL | | 0.63(0.39,1.00) |
| ATEZ+ChT | | 0.71(0.6,0.84) |
| NIVO | | 0.83(0.54,1.30) |
| IPIL+ChT | | 0.87(0.75,1.00) |
| ATEZ | | 1.1(0.68,1.90) |

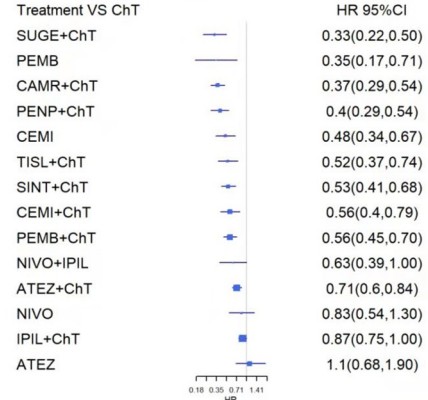

favours immunotherapy  favours chemotherapy

**Fig 3. Network meta-analysis (NMA) of immunotherapy compared with chemotherapy for PFS for overall study cohort, squamous and non-squamous cohort.** Summary estimates represent Hazard-Radio (HR) and 95% credibility intervals for PFS and OS. Interventions are ranked by Surface Under the Cumulative Ranking curve values. ATEZ, atelizumab; BEV, bevacizumab; CAMR, camrelizumab; CEMI, cemiplimab; CHT, chemotherapy; DURV, durvalumab; IPIL, ipilimumab; NIVO, nivolumab; PEMB, pembrolizumab; PENP, Penpulimab; SINT, sintilimab; SUGE, sugemalimab; TISL, Tislelizumab; TORI, Toripalimab; TREM, tremelimumab.

based CIT (P<0.05), however, no statistically significant difference was observed for PFS comparing with sintilimab, pembrolizumab, camrelizumab or tislelizumab based CIT, as well as pembrolizumab, cemiplimab monotherapy or nivolumab+ ipilimumab (P>0.05) (Fig 3 and S3 Table).

## PFS-NMA according to PD-L1 expression

**PD-L1 TPS > 50% cohort.** For patients with high PD-L1 TPS expression, all ICIs therapy except nivolumab monotherapy (HR, 0.62 [0.38–1.0]) were significantly more effective than chemotherapy. Camrelizumab based CIT has a 90.4% probability of being the best treatment for PFS in patients with high PD-L1 TPS expression, followed by sintilimab (82.3%) based CIT and ABCP (79.1%). All CIT regiments appear superior to ICI monotherapy for PFS in patients with high PD-L1 TPS expression; camrelizumab and sintilimab containing CIT were both significantly more effective than pembrolizumab or atezolizumab alone, and pembrolizumab + ipilimumab (P<0.05) (Fig 4).

**Intermediate PD-L1 TPS (1%-49%) cohort.** For patients with intermediate PD-L1 TPS expression, camrelizumab (HR, 0.46 [0.33–0.63]), cemiplimab (HR, 0.47 [0.33–0.67]), atezolizumab+bev (HR, 0.56 [0.41–0.77]), sintilimab (HR, 0.58 [0.41–0.82]), pembrolizumab (HR, 0.58 [0.44–0.78]), atezolizumab (HR, 0.61 [0.53–0.71]), or sugemalimab (HR, 0.71 [0.54–0.94]) based CIT as well as nivolumab/ipilimumab (HR, 0.62 [0.44–0.88]), atezolizumab monotherapy (HR, 0.72 [0.60–0.86]) were significantly better than chemotherapy. Camrelizumab containing CIT has an 89.5% probability of being the best treatment for PFS in patients with intermediate low PD-L1 TPS expression, followed by cemiplimab based CIT (86.7%) (S2 Fig).

**PD-L1 TPS < 1% cohort.** For patients with PD-L1 TPS of < 1%, pembrolizumab (HR, 0.51 [0.38–0.67]), camrelizumab (HR, 0.56 [0.42–0.74]), sintilimab (HR, 0.59 [0.43–0.81]), sugemalimab (HR, 0.66 [0.46–0.94]), tislelizumab (HR, 0.70 [0.49–1.0]), atezolizumab (HR, 0.72 [0.61–0.85]), or atezolizumab+bev (HR, 0.77 [0.61–0.98]) based CIT as well as nivolumab/ipilimumab (HR, 0.48 [0.27–0.85]) were significantly more effective than chemotherapy alone. Pembrolizumab containing CIT has an 84.2% probability of being the best treatment for PFS in patients with low PD-L1 TPS expression, and was significantly better than atezolizumab, atezolizumab+bev based CIT (P<0.05) (S3 Fig).

## Overall survival

**Short-term overall survival.** *OS-NMA for overall study cohort.* In the overall OS-NMA, the sintilimab (HR, 0.60 [0.42–0.86]), pembrolizumab (HR, 0.61 [0.52–0.71]), camrelizumab (HR, 0.64 [0.44–0.94]), sugemalimab (HR, 0.66 [0.45–0.98]), nivolumab+ipilimumab (HR, 0.69 [0.55–0.87]), cemiplimab (HR, 0.71 [0.54–0.94]), atezolizumab+bevacizumab (HR, 0.73 [0.63–0.84]), durvalumab+tremelimumab (HR, 0.77 [0.65–0.92]) and atezolizumab (HR, 0.81 [0.72–0.91]), based CIT all show a significant benefit in OS over chemotherapy. nivolumab/ipilimumab combined regiment (HR, 0.75 [0.66–0.86]) and pembrolizumab monotherapy (HR, 0.81 [0.71–0.93]) were also significantly more effective for OS than chemotherapy. Pembrolizumab containing CIT has an 89.5% probability to be the best regimen for OS, followed

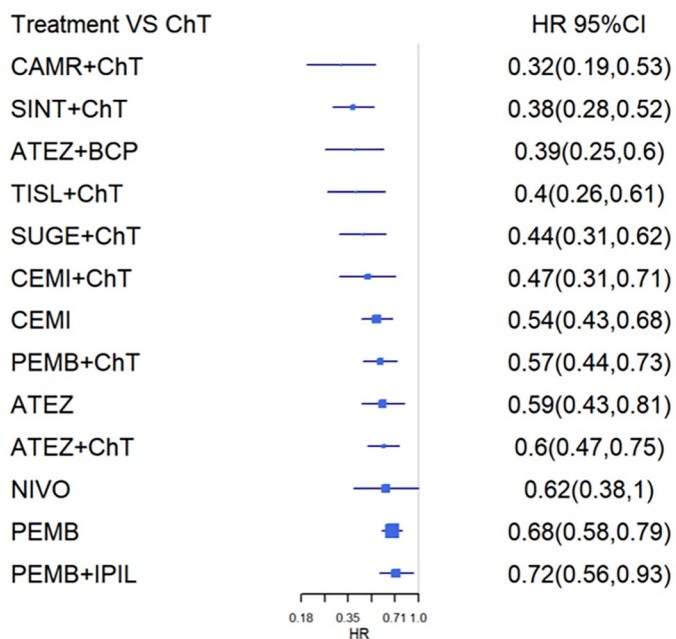

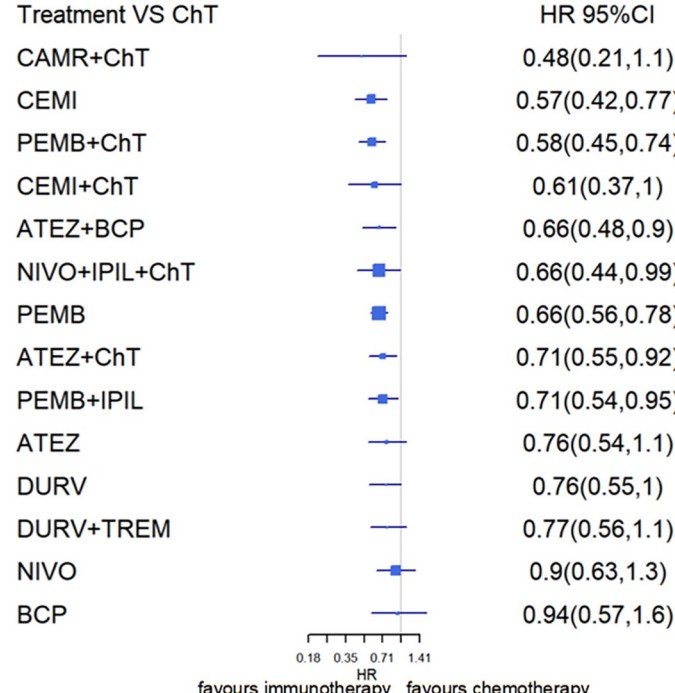

**Fig 4. Network meta-analysis (NMA) of immunotherapy compared with chemotherapy for PFS and OS for PD-L1 TPS > 50% cohort.** Summary estimates represent Hazard-Radio (HR) and 95% credibility intervals for PFS and OS. Interventions are ranked by Surface Under the Cumulative Ranking curve values. ATEZ, atelizumab; CAMR, camrelizumab; CEMI, cemiplimab; CHT, chemotherapy; DURV, durvalumab; IPIL, ipilimumab; NIVO, nivolumab; PEMB, pembrolizumab; SINT, sintilimab; SUGE, sugemalimab; TISL, Tislelizumab; TREM, tremelimumab.

by sintilimab containing CIT (84.7%), which two regimens were significantly more effective than atelizumab containing CIT, and pembrolizumab alone (P<0.05), but statistically significant difference wasn't observed when comparing with camrelizumab, cemiplimab, sugemalimab, sintilimab, atelizumab+bevacizumab, or nivolumab+ipilimumab plus chemotherapy (P >0.05) (Fig 5 and S2 Table).

*OS-NMA for squamous NSCLC.* In the squamous-histology OS-NMA, OS was significantly improved with the cemiplimab (HR, 0.56 [0.37–0.84]), camrelizumab (HR, 0.57 [0.34–0.96]), sintilimab (HR, 0.57 [0.35–0.92]), nivolumab+ipilimumab (HR, 0.62 [0.45–0.86]), and pembrolizumab (HR, 0.71 [0.58–0.87]), containing CIT. Cemiplimab and sintilimab based CIT showed numerical superiority over other CIT regimens for OS in patients with squamous NSCLC, with a 74.6% and 72.0% probability of being the best treatment respectively; but no statistically significant difference was observed comparing with pembrolizumab, camrelizumab, or nivolumab+ipilimumab containing CIT (Fig 5 and S4 Table).

*OS-NMA for non-squamous NSCLC.* In the non-squamous histology OS-NMA, pembrolizumab (HR, 0.50 [0.40–0.64]), atezolizumab+bevacizumab (HR, 0.67 [0.56–0.82]), nivolumab +ipilimumab (HR, 0.69 [0.55–0.87]), cemiplimab (HR, 0.79 [0.54–1.1]), or atezolizumab (HR, 0.80 [0.66–0.97]) combined platinum doublet were all significantly associated with OS benefit compared with chemotherapy alone. Pembrolizumab + chemotherapy had a 94.6% probability to be the best treatment regimen. Atezolizumab+bevacizumab -based CIT regimen was ranked as the second-best treatment, and a 67.0% probability to be the best regimen (Fig 5 and S4 Table).

*OS-NMA according to PS-L1 expression.* **PD-L1 TPS > 50% cohort.** For patients with high PD-L1 TPS expression, the pembrolizumab (HR, 0.58 [0.45–0.74]), atelizumab+bevacizumab (HR, 0.66 [0.48–0.90]), nivolumab+ipilimumab (HR, 0.66 [0.44–0.99]), atelizumab (HR, 0.71 [0.55–0.92]) based CIT, cemiplimab monotherapy (HR, 0.57 [0.42–0.77]) and pembrolizumab (HR, 0.66 [0.56–0.78]), as well as pembrolizumab+ipilimumab all show a significant benefit in OS over chemotherapy. Cemiplimab monotherapy showed numerical superiority over other CIT regimens for OS in patients with high PD-L1 TPS expression, with a 79.2% probability to be the best intervention for OS, followed by pembrolizumab (78.5%) and cemiplimab (67.2%) based CIT (Fig 4).

*OS-NMA for intermediate PD-L1 TPS (1%-49%) cohort.* For patients with intermediate PD-L1 TPS expression, cemiplimab (HR, 0.52 [0.32–0.84]), camrelizumab (HR, 0.52 [0.27– 1.0]), and pembrolizumab (HR, 0.59 [0.47–0.75]) based CIT showed numerical superiority over other ICI regimens, and were significantly more effective than chemotherapy. Cemiplimab based CIT has an 86.9% probability to be the best regimen for OS in patients with moderate PD-L1 TPS level, followed by camrelizumab (83.2%) and pembrolizumab (80.8%) based CIT (S2 Fig).

*OS-NMA for PD-L1 TPS < 1% cohort.* For patients with PD-L1 TPS of < 1%, camrelizumab (HR, 0.62 [0.41–0.94]), nivolumab+ipilimumab (HR, 0.62 [0.45–0.85]), pembrolizumab (HR, 0.65 [0.53–0.79]), atelizumab+bevacizumab (HR, 0.72 [0.58–0.89]), and atezolizumab (HR, 0.77 [0.69–0.87]) plus chemotherapy were significantly more effective than chemotherapy alone. Nivolumab+ipilimumab, camrelizumab, and pembrolizumab containing CIT show similar effectivity, with an 80.6%, 77.9% and 76.7% probability of being the best treatment in patients with low PD-L1 TPS expression respectively (S3 Fig).

**Long-term overal survival (≥24 months).** *OS-NMA for overall study cohort.* In the long-term (>2 years) OS-NMA, single ICI (atezolizumab, pembrolizumab, and durvalumab) CIT regiments (ABCP, atezolizumab, pembrolizumab, durvalumab, nivolumab+ipilimumab, and durvalumab+tremelimumab based CIT), and doublet ICIs (nivolumab/ipilimumab, and durvalumab/tremelimumab) were consistently associated with significant increased long-term OS

## OS for overall study cohort

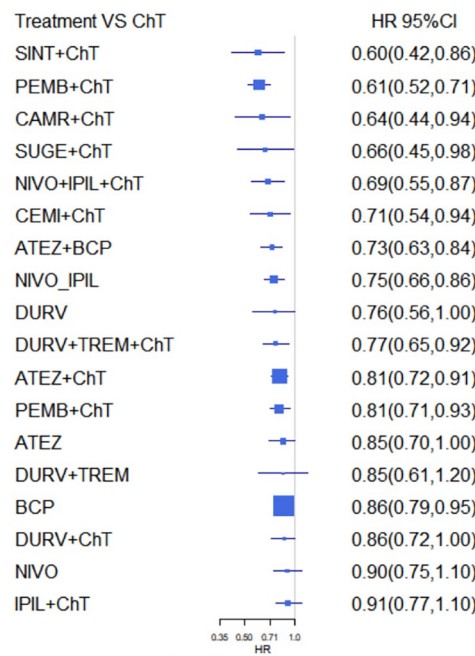

| Treatment VS ChT | | HR 95%CI |
|---|---|---|
| SINT+ChT | | 0.60(0.42,0.86) |
| PEMB+ChT | | 0.61(0.52,0.71) |
| CAMR+ChT | | 0.64(0.44,0.94) |
| SUGE+ChT | | 0.66(0.45,0.98) |
| NIVO+IPIL+ChT | | 0.69(0.55,0.87) |
| CEMI+ChT | | 0.71(0.54,0.94) |
| ATEZ+BCP | | 0.73(0.63,0.84) |
| NIVO_IPIL | | 0.75(0.66,0.86) |
| DURV | | 0.76(0.56,1.00) |
| DURV+TREM+ChT | | 0.77(0.65,0.92) |
| ATEZ+ChT | | 0.81(0.72,0.91) |
| PEMB+ChT | | 0.81(0.71,0.93) |
| ATEZ | | 0.85(0.70,1.00) |
| DURV+TREM | | 0.85(0.61,1.20) |
| BCP | | 0.86(0.79,0.95) |
| DURV+ChT | | 0.86(0.72,1.00) |
| NIVO | | 0.90(0.75,1.10) |
| IPIL+ChT | | 0.91(0.77,1.10) |

## OS for non-squamous NSCLC

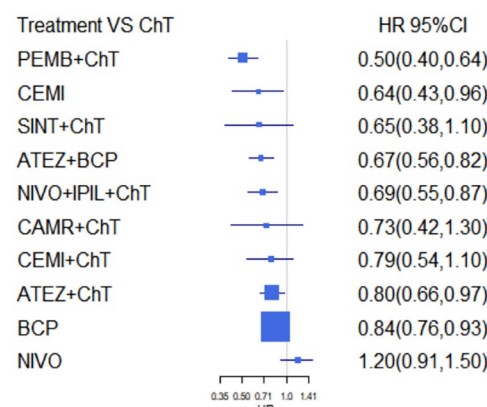

| Treatment VS ChT | | HR 95%CI |
|---|---|---|
| PEMB+ChT | | 0.50(0.40,0.64) |
| CEMI | | 0.64(0.43,0.96) |
| SINT+ChT | | 0.65(0.38,1.10) |
| ATEZ+BCP | | 0.67(0.56,0.82) |
| NIVO+IPIL+ChT | | 0.69(0.55,0.87) |
| CAMR+ChT | | 0.73(0.42,1.30) |
| CEMI+ChT | | 0.79(0.54,1.10) |
| ATEZ+ChT | | 0.80(0.66,0.97) |
| BCP | | 0.84(0.76,0.93) |
| NIVO | | 1.20(0.91,1.50) |

## OS for squamous NSCLC

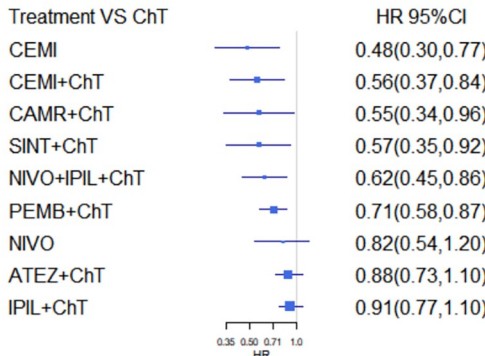

| Treatment VS ChT | | HR 95%CI |
|---|---|---|
| CEMI | | 0.48(0.30,0.77) |
| CEMI+ChT | | 0.56(0.37,0.84) |
| CAMR+ChT | | 0.55(0.34,0.96) |
| SINT+ChT | | 0.57(0.35,0.92) |
| NIVO+IPIL+ChT | | 0.62(0.45,0.86) |
| PEMB+ChT | | 0.71(0.58,0.87) |
| NIVO | | 0.82(0.54,1.20) |
| ATEZ+ChT | | 0.88(0.73,1.10) |
| IPIL+ChT | | 0.91(0.77,1.10) |

**Fig 5. Network meta-analysis (NMA) of immunotherapy compared with chemotherapy for OS for overall study cohort, squamous and non-squamous cohort.** Summary estimates represent Hazard-Radio (HR) and 95% credibility intervals for PFS and OS. Interventions are ranked by Surface Under the Cumulative Ranking curve values. ATEZ, atelizumab; BEV, bevacizumab; CAMR, camrelizumab; CEMI, cemiplimab; CHT, chemotherapy; DURV, durvalumab; IPIL, ipilimumab; NIVO, nivolumab; PEMB, pembrolizumab; PENP, Penpulimab; SINT, sintilimab; SUGE, sugemalimab; TISL, Tislelizumab; TORI, Toripalimab; TREM, tremelimumab.

rate compared with chemotherapy alone and BEV plus chemotherapy. Durvalumab mono-therapy (HR, 0.48 [0.32–0.72]) and nivolumab/ipilimumab (HR, 0.49 [0.38–0.64]) showed numerical superiority over other ICI regimens for long-term OS, with a 79.9% and a 79.8% probability of being the best treatment respectively, followed by pembrolizumab monotherapy (74.8%) and ABCP (70.9%).

## Objective response rate

In the overall NMA analysis of ORR, all CIT regiments except ipilimumab containing CIT sig-nificantly improved ORR compared with chemotherapy. The CIT regiments were superior to single-agent ICI (atezolizumab, pembrolizumab and nivolumab) for ORR. Pembrolizumab containing CIT showed numerical superiority over other ICI regimens for ORR, with a 92.2% probability of being the best treatment, followed by penpulimab (89.1%) and sugemalimab (78.0%) based CIT (S4 Fig).

In the squamous NSCLC NMA of ORR, ORR was significantly improved with the tislelizu-mab (MD, 23.0 [11.0–35.0]), pembrolizumab (MD, 24.0 [16.0–32.0]), penpulimab (MD, 26.0 [16.0–37.0]), and camrelizumab (MD, 28.0 [18.0–38.0]) containing CIT. Camrelizumab con-taining CIT showed numerical superiority over other CIT regimens for ORR in patients with squamous NSCLC, with an 85.6% probability of being the best treatment, followed by penpuli-mab (80.0%) and pembrolizumab (73.1%) based CIT.

In the non-squamous histology ORR-NMA, camrelizumab (MD, 13.0 [5.7–21.0]), atezoli-zumab (MD, 16.0 [11.0–22.0]), tislelizumab (MD, 20.0 [9.1–32.0]), sintilimab (MD, 22.0 [12.0–32.0]), or pembrolizumab (MD, 28.0 [21.0–35.0]), combined platinum doublet was all significantly associated with ORR benefit compared with chemotherapy alone. Pembrolizu-mab + chemotherapy had a median rank of being the best treatment regimen for ORR in patients with non-squamous NSCLC, and was associated with a 93.8% probability of being the best treatment.

## Safety

In the OR-NMAs for grade 3–5 TRAEs, all single-agent ICI and doublet ICIs regiments exhibit markedly lower odds of 3–5 TRAEs compared to CIT regiments and chemotherapy alone. Nivolumab was shown to be the safest therapy among the regimens evaluated (OR 0.21, 95% CI 0.14–0.31), followed by cemiplimab (OR 0.22, 95% CI 0.15–0.32) and pembrolizumab alone (OR 0.33, 95% CI 0.14–0.31). Compared with chemotherapy alone, grade 3–5 TRAEs were significantly higher for ipilimumab (OR 2.0, 95% CI 1.5–2.7), camrelizumab (OR 1.7, 95% CI 1.3–2.4), cemiplimab (OR 1.7, 95% CI 1.1–2.6), and nivolumab+ipilimumab (OR 1.5, 95% CI 1.1–2.0) based CIT as well as ABCP (OR 1.4, 95% CI 1.1–1.9).

## Discussion

This NMA is based on 32 first-line RCTs for advanced NSCLC, which involving 18,656 patients randomized to 22 ICI-based regimens with 13 different ICI agents, and including recent published data of long-term follow-up of immunotherapy trials. The present analysis is

substantially more comprehensive than previous meta-analysis for first-line ICI therapy for advanced NSCLC [2, 3, 22, 23]. The much larger evidence base, obtained through exhaustive search for published article and abstracts from recent major conference proceedings of the ASCO, EMSO, and WCLC. The main findings of this study including: (1) A series of ICI regiments (including ICI plus chemotherapy, doublet ICIs, doublet ICIs plus chemotherapy) emerged, and showed significant PFS and OS benefit than chemotherapy and chemotherapy +BEV for advanced wild-type NSCLC. (2) In comprehensive terms of PFS, CIT were significantly more effective than ICI monotherapy and doublet ICIs, penpulimab based CIT and ABCP showed numerical superiority over other ICI regimens for PFS. (3) In terms of OS for patients with non-squamous NSCLC, pembrolizumab containing CIT ranked to be the best treatment regimen, and followed by sintilimab based CIT; while for OS in patients with squamous NSCLC, camrelizumab and sintilimab based CIT were the most effective regimens. Cemiplimab monotherapy showed numerical superiority over other CIT regimens for OS in patients with high PD-L1 TPS expression. (4) No statistically significant benefit for OS or PFS was observed for doublet ICIs based therapies comparing with single ICI based therapies. (5) For >24 months follow-up, the atezolizumab, pembrolizumab, nivolumab and durvalumab containing ICI therapy (including monotherapy, CIT, doublet ICIs, doublet ICIs plus chemotherapy) all provide a durable, clinically meaningful long-term OS benefit over chemotherapy and BEV+ chemotherapy.

For overall study cohort, we observed that CIT regiments were overall superior to ICI monotherapy and doublet ICIs in terms of PFS and ORR. The PFS benefit of a series of CIT regiments including penpulimab, camrelizumab, pembrolizumab, sugemalimab, tislelizumab, toripalimab, sintilimab based CIT and ABCP is comparable for overall population. In the OS NMA for overall study population, pembrolizumab containing CIT ranks being the best treatment, followed by sintilimab containing CIT. These two CIT regimens were also statistically superior to ICI monotherapy (atelizumab or pembrolizumab) and atelizumab based CIT, but not statistically superior to camrelizumab, cemiplimab, sugemalimab, atelizumab+- bevacizumab, or nivolumab+ipilimumab based CIT. The similar PFS and OS benefits achieved for the above CIT regiments suggest new better practice-changing options for advanced NSCLC without sensitizing EGFR or ALK genetic alterations. Additional data including cost-effectiveness and toxicity of ICIs regiments need to be considered to formulate national and health policy decisions.

We found that the majority of the ICIs therapies were more efficacious compared with chemotherapy (Table 2). However, some ICIs monotherapy and CIT regiments don't provide

**Table 2. The optimal treatment regimens for NSCLC according to PD-L1 expression level and different pathological types.**

| | OS | | PFS | |
|---|---|---|---|---|
| | **Best regimen** | **Rank probability (%)** | **Best regimen** | **Rank probability (%)** |
| PD-L1 TPS expression | | | | |
| < 1% | NIVO+IPIL+ChT | 80.6 | PEMB+ChT | 84.2 |
| 1–49% | CEMI+ChT | 86.9 | CAMR+ChT | 89.5 |
| >50% | CEMI | 79.2 | CAMR+ChT | 90.4 |
| Squamous | CEMI+ChT | 74.6 | SUGE+ChT | 91.2 |
| Non-squamous | PEMB+ChT | 94.6 | ATEZ+BCP | 92.3 |
| Overall | PEMB+ChT | 89.5 | PENP+ChT | 95.7 |

Abbreviation: ATEZ, atelizumab; BEV, bevacizumab; CAMR, camrelizumab; CEMI, cemiplimab; ChT, chemotherapy; IPIL, ipilimumab; NIVO, nivolumab; PEMB, pembrolizumab; PENP, Penpulimab; SUGE, sugemalimab.

statistically significant PFS or OS benefit over chemotherapy. Durvalumab plus tremelimumab or durvalumab monotherapy did not significantly improved OS or PFS vs chemotherapy in patients with NSCLC and PD-L1 TC $\geq$ 25% [24]. Similarly, nivolumab monotherapy wasn't significantly improved PFS compared with chemotherapy for patients with advanced or recurrent NSCLC whose PD-L1 expression > 5%, and OS benefit was also negative [25]. Although PFS and OS was statistically significantly improved with first-line nivolumab plus ipilimumab than chemotherapy, the PFS and OS benefit from this doublet ICIs regiment is relative inferior to some other ICI treatment examined. ACP regiment also provide less benefit in PFS or OS compared with ABCP, camrelizumab, pembrolizumab and sintilimab containing CIT, with lower SUCRA value in PFS and OS NMA. The inferior efficacy, high cost of doublet ICIs, and toxicity making these strategies less favourable options.

For patients with squamous NSCLC, the PFS NMA analysis suggest sugemalimab and camrelizumab based CIT was ranked as the first- and second- best treatment option, which were statistically superior to pembrolizumab and atezolizumab based CIT, and with similar efficacy with penpulimab, cemiplimab, sintilimab, or tislelizumab based CIT. In the OS NMA of squamous subgroup, camrelizumab and cemiplimab based CIT ranked as the first- and second-best therapy, followed by sintilimab based CIT. The OS benefit of these three CIT regimens were numerical superior to pembrolizumab containing CIT. The OS data is byfar immature for sugemalimab, penpulimab, and tislelizumab based CIT [25]. As OS should be considered as the standard measure of clinical benefit for cancer patients, the present NMA results suggest that camrelizumab, cemiplimab and sintilimab based CIT may be recommended as an initial first-line intervention for those with advanced, squamous NSCLC.

For nonsquamous subgroup, ABCP with the highest probability to be the best treatment in PFS NMA, and followed by sintilimab, pembrolizumab based CIT. However, the PFS benefit of ABCP did not translate into the best OS benefit. Pembrolizumab containing CIT has the highest probability (94.6%) to be the best ICI regimen for OS in patients with advanced, non-squamous NSCLC. Atezolizumab+bevacizumab- based CIT regimen ranked second for OS benefit, whereas the mean rank (67.0%) is substantially lower than pembrolizumab based CIT. ABCP is numercialy inferior to pembrolizumab and sintilimab based CIT in term of survival benefit. The results were generally consistent with with previous network meta-analysis by Frederickson et al., which study showed pembrolizumab plus platinum- based chemotherapy had statistically significant OS benefit compared with other interventions in advanced nonsquamous NSCLC patients without sensitive oncogenic drivers [23].

For PD-L1-high patients with NSCLC, CIT regiments exhibit superior PFS benefit compared with single agent ICI or doublet ICIs. Camrelizumab based CIT ranks first for PFS benefit, followed by sintilimab based CIT and ABCP. Whereas, the best survival benefit was observed for cemiplimab monotherapy, which performed marginally better than pembrolizumab containing CIT. Both of cemiplimab monotherapy and pembrolizumab containing CIT are numerical superior to pembrolizumab alone, cemiplimab based CIT and ABCP. Atezolizumab, nivolumab, and durvalumab alone didn't significantly improved OS for PD-L1-high patients with NSCLC. Findings from a previous network meta-analysis by Liu et al. showed for patients with PD-L1-high expression, CIT regiments should be superior to pembrolizumab alone. However, based on the recent additions to the treatment space of advanced NSCLC, the present NMA suggests cemiplimab monotherapy should also be recommended as the initial treatment choice for PD-L1-high patients.

In the PD-L1 negative and intermediate cohort, CIT regiments exhibit superior PFS and OS benefit to ICI alone. Cemiplimab based CIT has a mean rank of being the best treatment for OS in patients with intermediate PD-L1 TPS expression, followed by camrelizumab and pembrolizumab based CIT, without an apparent disparity between the three combinations.

Nivolumab+ipilimumab, camrelizumab, and pembrolizumab containing CIT show similar superior effectivity for patients with negative PD-L1 expression. This evidence suggests that camrelizumab, pembrolizumab based CIT generally be preferred first-line treatments for PD L1-intermediate or negative advanced NSCLC patients; cemiplimab based CIT may also be recommended as a preferred first-line treatment for PD L1-intermediate cohort and Nivolumab+ipilimumab plus chemotherapy for PD-L1 negative patients.

## Long-term

After more than 2 years follow-up, all ICI regiments were consistently resulted in substantially increased long-term OS and PFS rate compared with chemotherapy alone and BEV plus chemotherapy. Substantial long-term survival benefit was observed across single ICI, CIT, and doublet ICIs regiments. Specially, single ICI and doublet ICIs regiments showed numerical superiority over CIT regimens. Although durvalumab monotherapy didn't provide statistically significant median OS or PFS benefit over chemotherapy, durvalumab has a mean rank of being the best treatment for long-term OS benefit, followed by Nivolumab+ipilimumab and pembrolizumab monotherapy. There is an ongoing debate about which is the most adequate primary endpoint in clinical trial for NSCLC. The current NMA results suggested that median OS or PFS is generally incomplete in settings for evaluating the treatment effects of ICI regiments for NSCLC. Previous studies on NSCLC showed that an increase in PFS does not necessarily result in OS benefit; however, post-progression survival is strongly associated with OS after early-line treatment. Sustained survival benefit from ICI therapy suggests that both median OS and post-progression survival have increased along the years, which illustrated the delayed treatment effect of ICI therapy. Moreover, first-line pembrolizumab monotherapy and pembrolizumab based CIT were shown to improve PFS-2, and the PFS-2 was nearly doubled for patients in the pembrolizumab based CIT group. The outcome of PFS-2 suggests that despite the high crossover rate, treatment benefit of the first line therapy was maintained into the next line. As a result, post-progression survival and PFS-2 should also be considered as endpoint in clinical trials of ICI therapy for NSLCL. Substantial long-term survival benefit suggests target and discover specific patients who respond best to ICI treatment has become the major issue. However, despite the PD-L1 expression was approved as a standard biomarker in the setting of ICI treatment for advanced NSCLC, the response prediction was imperfect. There are other potential predictive biomarkers which could be factored into identify populations who respond best to specific combinations. Further research is required to identify additional biomarkers, which will help to align specific ICI regiments to specific patient groups.

The literature search for the present NMA was comprehensive, with the largest number of phases III RCTs for first-line immunotherapy treatments for advanced NSCLC, including 15748 patients enrolled in 32 studies. The 32 studies included recent published trials, long-term update of previous trials and abstracts from 2021 major conference proceedings to date. A previous NMA by Liu et al., based on one phase II and nine phase III studies, involving 6,124 patients with metastatic NSCLC, shows CIT intervention is superior to pembrolizumab monotherapy for patients with PD-L1 level $\geq$ 1% and especially for those with PD-L1 $\geq$ 50%. For non-squamous NSCLC, Bev +chemotherapy should also be recommended as an initial treatment for patients with PDL1 $\geq$ 1%, since Bev +chemotherapy was not inferior to Pembrolizumab alone. However, with incorporated a considerable amount of recent published data, the present NMA shows the best survival benefit for PD-L1 $\geq$ 50% cohort was observed for cemiplimab monotherapy, which performed marginally better than pembrolizumab containing CIT. For non-squamous NSCLC, the present NMA suggest all ICI regiments show long-term OS benefit compared to Bev +chemotherapy. A recent NMA was conducted in 2021 by

Herbst et al., which including 17 clinical trials, limited on patients with stage IV NSCLC and high PD-L1 expression subgroup. Whereas, our NMA explored the relative benefit of ICI regimens on OS/PFS by PD-L1 levels and irrespective of PD-L1-positivity.

Our review has some limitations. First, most of the recent published data with short duration of follow-up, the number of OS events was not mature and the long-term PFS and OS rates were not available. The short follow-up prevented the full assessment of specific ICI regiment. Second, the majority studies included implement the 22C3 pharmDx assay to test the PD-L1 TPS, which is defined based on the proportion of tumor cells in membranous PD-L1 staining.38 PD-L1 positivity refer to a tumour proportion score of 1% or higher. Whereas, IMpower studies utilize SP142 assaying to test PD-L1 level [15, 17, 40]. Different assay method and reference value might lead to misclassification, affecting outcome assessment. Third, the different disease stage of study cohort may also be a confounding factor. Keynote, IMpower, and CheckMate studies enrolled stage IV NSCLC, whereas others included patients with stage IIIB or IIIC disease, meaning that the comparison need to be interpreted with great caution. Fouth, we lacked data from head-to-head comparisons of these ICIs regiments, and many conclusions are reliant on indirect comparisons. The limited value of indirect comparisons of results obtained from multiple different regimens, with varying follow-up time, suggests the results need to be taken with caution. Last, non-Asian patients were not recruited in the ORIENT, RATIONALE, Camel and CHOICE studies, and the results limited to Chinese subpopulation and therefore should be interpreted with caution.

Notwithstanding these limitations, the findings from this network meta-analysis represent the most comprehensive currently available evidence base for initial choice about first-line immunotherapies for advanced NSCLC patients without known EGFR mutations or ALK translocations. However, the findings from comparisons among ICI regiments should be interpreted by the potential limitations of the study design, and the heterogenous patient populations. The results of the present NMA might suggest or provide basis for first-line ICI therapy decision for advanced NSCLC patients without oncogenic driver mutations.

## Supporting information

**S1 Checklist. PRISMA 2020 checklist.**
(DOCX)

**S1 Table. Outcome data were extracted from 32 first-line phase III RCTS.**
(XLSX)

**S2 Table. PFS, and OS comparative profiles for overall study cohort according to network meta-analysis (NMA).**
(DOC)

**S3 Table. PFS comparative profiles for squamous and non-squamous cohort according to network meta-analysis (NMA).**
(DOC)

**S4 Table. OS comparative profiles for squamous and non-squamous cohort according to network meta-analysis (NMA).**
(DOC)

**S1 Fig. Network meta-analysis (NMA) of eligible comparisons for ORR.** The size of the circle corresponds to to the number of randomly assigned participants (the circle of the chemotherapy group proportional to the number of participants divided by 6). The directly compared interventions are linked with a line, and the thickness of the lines is proportional to

the number of trials that assessed the comparison.
(TIF)

**S2 Fig. Network meta-analysis (NMA) of immunotherapy compared with chemotherapy for PFS and OS for PD-L1 TPS 1–49% cohort.** Summary estimates represent Hazard-Radio (HR) and 95% credibility intervals for PFS and OS. Interventions are ranked by Surface Under the Cumulative Ranking curve values.
(TIF)

**S3 Fig. Network meta-analysis (NMA) of immunotherapy compared with chemotherapy for PFS and OS for PD-L1 TPS < 1% cohort.** Summary estimates represent Hazard-Radio (HR) and 95% credibility intervals for PFS and OS. Interventions are ranked by Surface Under the Cumulative Ranking curve values.
(TIF)

**S4 Fig. ORR comparative profiles for overall study cohort according to network meta-analysis (NMA).** Each cell contains the Mean difference (MD) and 95% credibility intervals for PFS and OS; significant results are emboldened.
(TIF)

## Author Contributions

**Conceptualization:** Tingting Liu, Weiwei Fang, Guiye Qi.

**Data curation:** Tingting Liu, Sijia Wu, Weiwei Fang, Hongkai Li, Yao Liu.

**Formal analysis:** Sijia Wu, Weiwei Fang, Hongkai Li.

**Investigation:** Sijia Wu.

**Methodology:** Hongkai Li, Lili Su, Guiye Qi, Huaichen Li.

**Project administration:** Weiwei Fang.

**Resources:** Hongkai Li, Guiye Qi.

**Software:** Hongkai Li.

**Supervision:** Lili Su, Huaichen Li, Yao Liu.

**Validation:** Lili Su, Huaichen Li.

**Visualization:** Huaichen Li, Yao Liu.

**Writing – original draft:** Tingting Liu, Yao Liu.

**Writing – review & editing:** Huaichen Li, Yao Liu.

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
