## [Decision Letter · Decision Letter 0]

24 Jan 2023

PONE-D-22-24716Identifying optimal first-line immune checkpoint inhibitors

based regiment s for advanced non-small cell lung cancer without oncogenic drivers: a systematic review and network meta-analysisPLOS ONE

Dear Dr. Liu,

Thank you for submitting your manuscript to PLOS ONE. After careful consideration, we feel that it has merit but does not fully meet PLOS ONE’s publication criteria as it currently stands. Therefore, we invite you to submit a revised version of the manuscript that addresses the points raised during the review process.

We look forward to receiving your revised manuscript.

Kind regards,

Rama Krishna Kancha

Academic Editor

PLOS ONE

Journal Requirements:

 This work was supported by Natural Science Foundation of Shandong Province (No. ZR2021MH006), and the corresponding author doctor LY received this award.

6. Please ensure that you refer to Figure 3-5 in your text as, if accepted, production will need this reference to link the reader to the figure.

7. Please include your tables as part of your main manuscript and remove the individual files. Please note that supplementary tables (should remain/ be uploaded) as separate "supporting information" files.

Additional Editor Comments:

The reviewers opined that the study is valuable except for few suggestions that include (1) the importance of this study is overstated saying that it influences the decision regarding first-line treatment and (2) concerns regarding certain parameters used in this study. Please address reviewers' concerns point-wise in the rebuttal.

Reviewers' comments:

Reviewer's Responses to Questions

**Comments to the Author**

1. Is the manuscript technically sound, and do the data support the conclusions?

Reviewer #1: Yes

Reviewer #2: Yes

2. Has the statistical analysis been performed appropriately and rigorously? 

Reviewer #1: Yes

Reviewer #2: I Don't Know

3. Have the authors made all data underlying the findings in their manuscript fully available?

Reviewer #1: Yes

Reviewer #2: Yes

4. Is the manuscript presented in an intelligible fashion and written in standard English?

Reviewer #1: Yes

Reviewer #2: No

5. Review Comments to the Author

Reviewer #1: The manuscript's topic if of great interest and relevance. Also the aim and outcomes are well described. The methodogy is correct. The presentation of the results is exhaustive and clear. I have just few minor comments that require revisions:

Comments:

1) In the conclusion of the abstract and of the full text, the authors state that their findings may guide the choice of the first-line treatment of NSCLC. In my opinion, even if comprehensive and weel-designed, a meta-analysis cannot directly led to prefer a treatment over another. It can suggest or give basis for a clinical-based evidence therapeutic decision. Please changes the terms of this sentence both in the abstract as well as in the conclusions.

2) at the end of page 9 when presenting the aim of the study ".to determine the optimal first-line immunotherapies ......", I would prefer the use of the following terms "to compare the different first-lines regimens in term of efficacy" (please ad the parameters chosen for comparison).

Reviewer #2: Immune checkpoint inhibitors (ICIs) have greatly changed the treatment pattern of advanced and metastatic NSCLC. In recent years, the results of a number of prospective studies evaluating the efficacy and safety of immunochemotherapy in the first-line treatment of advanced NSCLC have been published. The authors undertook the task of summarising the results of these studies in the form of a network meta-analysis with the aim of determining the optimal first-line immunotherapies for advanced NSCLC. The topic addressed is important from a practical point of view. However, I have a few comments.

1. In the discussion, the authors highlight the limitations of the manuscript - for example, related to the ethnic diversity of the populations eligible for treatment or the diagnostic issues of PD-L1. They also highlight the lack of studies directly comparing different immunochemotherapy regimens. However, it will be worth much more emphasizing the limited value of indirect comparisons of results obtained by patients treated across multiple different regimens, with varying follow-up times and different population sizes.

2. The authors conclude that the results of the present NMA may assist the initial choice of first-line immunotherapies for advanced NSCLC patients. In my opinion, one should be tempted to provide further clarification, a summary, and a proposed treatment algorithm included in the table. Specifying the following subpopulations: PD-L1 <1%, 1-49%, >50% in each histological type (squamous vs non-squamous).

3. In the 'data extraction' section, it is stated that among the variables collected were the age and gender of the patients. Were analyses of the prognostic value of these parameters performed?

4. My additional suggestion is to add the word „mutations“ in the title and further parts of the manuscript (oncogenic driver mutations). Furthermore, it is worth noting, that the paper only refers to abnormalities in the EGFR and ALK genes. Therefore, maybe consider specifying in the title which molecular abnormalities are referring to.

5. Furthermore- please note that numerous items of the references need to be corrected.

6. PLOS authors have the option to publish the peer review history of their article (what does this mean?). If published, this will include your full peer review and any attached files.

Reviewer #1: No

Reviewer #2: No

---

## [Author Response · Author response to Decision Letter 0]

13 Mar 2023

Dear editors

Thanks you very much for your comments and suggestions. 

We have revised the manuscript, according to the comments and suggestions of reviewers and editor, and responded, point by point to, the comments as listed below. We have highlighted all the changes in red.

I would like to re-submit this revised manuscript to Plos One, and hope it is acceptable for publication in the journal.

Looking forward to hearing from you soon.

With kindest regards,

Yours Sincerely 

Yao Liu

Replies to Reviewers and Editor

First of all, we thank both reviewers and editor for their positive and constructive comments and suggestions.

Reviewer #1: The manuscript's topic if of great interest and relevance. Also the aim and outcomes are well described. The methodogy is correct. The presentation of the results is exhaustive and clear. I have just few minor comments that require revisions:

Comments:

1) In the conclusion of the abstract and of the full text, the authors state that their findings may guide the choice of the first-line treatment of NSCLC. In my opinion, even if comprehensive and weel-designed, a meta-analysis cannot directly led to prefer a treatment over another. It can suggest or give basis for a clinical-based evidence therapeutic decision. Please changes the terms of this sentence both in the abstract as well as in the conclusions.

Answer: This is a very constructive suggestion. We have rewritten these sentences both in the abstract as well as in the conclusions.

2) at the end of page 9 when presenting the aim of the study ".to determine the optimal first-line immunotherapies ......", I would prefer the use of the following terms "to compare the different first-lines regimens in term of efficacy" (please ad the parameters chosen for comparison).

Answer: Thank you for the suggestion. We have rewritten this sentence in the paragraph above the Methods section.

Reviewer #2: Immune checkpoint inhibitors (ICIs) have greatly changed the treatment pattern of advanced and metastatic NSCLC. In recent years, the results of a number of prospective studies evaluating the efficacy and safety of immunochemotherapy in the first-line treatment of advanced NSCLC have been published. The authors undertook the task of summarising the results of these studies in the form of a network meta-analysis with the aim of determining the optimal first-line immunotherapies for advanced NSCLC. The topic addressed is important from a practical point of view. However, I have a few comments.

1. In the discussion, the authors highlight the limitations of the manuscript - for example, related to the ethnic diversity of the populations eligible for treatment or the diagnostic issues of PD-L1. They also highlight the lack of studies directly comparing different immunochemotherapy regimens. However, it will be worth much more emphasizing the limited value of indirect comparisons of results obtained by patients treated across multiple different regimens, with varying follow-up times and different population sizes.

Answer: Thank you for the constructive suggestion. We have added the limited value of indirect comparisons of results in the discussion.

2. The authors conclude that the results of the present NMA may assist the initial choice of first-line immunotherapies for advanced NSCLC patients. In my opinion, one should be tempted to provide further clarification, a summary, and a proposed treatment algorithm included in the table. Specifying the following subpopulations: PD-L1 <1%, 1-49%, >50% in each histological type (squamous vs non-squamous).

Answer: Thank you for the constructive suggestions. We have added a table (Table 2) in the revised manuscript, which further summarize the optimal treatment regimens for NSCLC according to PD-L1 expression level and different pathological types.

3. In the 'data extraction' section, it is stated that among the variables collected were the age and gender of the patients. Were analyses of the prognostic value of these parameters performed?

Answer: We regret that we were unable to conduct additional analyses of the prognostic value of these parameters.

4. My additional suggestion is to add the word “mutations” in the title and further parts of the manuscript (oncogenic driver mutations). Furthermore, it is worth noting, that the paper only refers to abnormalities in the EGFR and ALK genes. Therefore, maybe consider specifying in the title which molecular abnormalities are referring to.

Answer: Thank you for the suggestion. We have added the word “mutations” in the title and further parts of the manuscript.

5. Furthermore- please note that numerous items of the references need to be corrected.

Answer: We are very sorry for these mistakes. We have corrected the references (ref 4, ref 14, ref 21, ref 36 and ref 41) in the Reference section of the revised manuscript. In addition, some data were extracted from the contents of the major conferences and have not been published as articles, for which we may not modify these references.

Furthermore, we revised the manuscript according to the Journal Requirements section. The changes as listed below.

1. We have added the Role of Funder statement section and include this amended section in our manuscript and cover letter.

2. We have rectified the Data Availability Statement section. The relevant data are within the supporting files.

3. We have moved ethics statement to the Methods section.

4.We have modified the position of figures and tables in our text.

5. According to the comments, we have inserted the tables into the manuscript.

6. Tingting Liu, Sijia Wu, and Weiwei Fang should be considered co-first authors. We are very sorry for this mistake, and we have rectified that section in our manuscript.

---

## [Editor Report · Decision Letter 1]

15 Mar 2023

Identifying optimal first-line immune checkpoint inhibitors based regiments for advanced non-small cell lung cancer without oncogenic driver mutations: a systematic review and network meta-analysis

PONE-D-22-24716R1

Dear Dr. Liu,

We’re pleased to inform you that your manuscript has been judged scientifically suitable for publication and will be formally accepted for publication once it meets all outstanding technical requirements.

Kind regards,

Rama Krishna Kancha

Academic Editor

PLOS ONE
---

## [Editor Report · Acceptance letter]

3 Apr 2023

PONE-D-22-24716R1 

Identifying optimal first-line immune checkpoint inhibitors based regiments for advanced non-small cell lung cancer without oncogenic driver mutations: a systematic review and network meta-analysis 

Dear Dr. Liu:

I'm pleased to inform you that your manuscript has been deemed suitable for publication in PLOS ONE. Congratulations! Your manuscript is now with our production department. 

Kind regards, 

on behalf of

Dr. Rama Krishna Kancha 

Academic Editor

PLOS ONE